Cultural adaptation; youth of color; adolescent mental health; community mental health; global mental health

**Corresponding authors:**
Adam D. Brown and Janus Wong;
Emails: brownad@newschool.edu;
januswon@usc.edu

# Community-based adaptation of early adolescent skills for emotions for urban adolescents and caregivers in New York City

Janus Wong[1,2] , Tina Xu[1], Cheenar Shah[1,3], Liam Miccoli[1], Josheka Chauhan[1], Nora Garbuno Inigo[1], Kendall Pfeffer[1,4], Dana Ergas Slachevsky[1], Arian Holman[1], Eva Wong[5], Heather Day[5], Kala Ganesh[5], Eliot Assoudeh[5], Brandon A. Kohrt[3] and Adam D. Brown[1,6]

[1]Center for Global Mental Health, Department of Psychology, The New School for Social Research, New York, NY, USA; [2]Department of Psychology, University of Southern California, Los Angeles, CA, USA; [3]Center for Global Mental Health Equity, Department of Psychiatry and Behavioral Health, George Washington University, Washington, DC, USA; [4]Icahn School of Medicine, Mount Sinai Hospital; [5]Office of Community Mental Health, New York City Mayor, New York, NY, USA and [6]Department of Psychiatry, New York University Grossman School of Medicine

## Abstract

An increasing number of studies have sought to explore the applicability of scalable mental health interventions to bridge the adolescent mental health treatment gap. This study aimed to adapt the World Health Organization's mental health intervention Early Adolescent Skills for Emotion (EASE) for urban communities in New York City (NYC). Following the mental health Cultural Adaptation and Contextualization for Implementation framework and in collaboration with three Brooklyn community-based organizations and the NYC Mayor's Office of Community Mental Health, the intervention was intensively workshopped through eight weekly sessions with adolescents ($n = 18$) and caregivers ($n = 12$). Documentation of the process followed the Reporting Cultural Adaptation in Psychological Trials criteria. Surface adaptations involved revising the storybook to reflect key challenges faced by adolescents and caregivers of these communities, such as social media usage, economic stressors, and racial diversity. Deep adaptations addressed cultural concepts of distress by incorporating topics such as identity exploration, socioemotional learning, and the mind–body connection. Feedback from stakeholders indicated that the basic components of EASE are relevant for members in their communities, but additional changes would foster greater engagement and community building. These findings will inform upcoming program implementation across NYC and may guide adaptation work in other contexts.

## Impact statement

As global strategies seeking to address the mental health services gap through scalable mental health interventions become more widely adopted, there is a growing importance for cultural adaptation of psychosocial interventions to reflect local needs and contexts. This study, which reports on the cultural adaptation process of a World Health Organization (WHO) mental health intervention for youth and caregivers, Early Adolescent Skills for Emotions (EASE), is the first step towards implementing EASE in the United States and expanding mental health services for New York City (NYC) adolescents in economically marginalized communities of color. Through documenting the researchers' close partnership with the local government, youth and caregivers, the study highlights the importance of centering community voices in intervention development, implementation, and dissemination. It also indicates the necessity of culturally sensitive approaches to co-create strategies that are acceptable, relevant, and engaging for underrepresented groups. The study's findings are being used to inform the pilot implementation and evaluation of EASE in NYC in 2025. It may also guide community-based adaptation work in other contexts globally. Additionally, the study offers insights into how a novel strategy focused on community building and social and emotional learning may increase access to care for adolescents in need of mental health support.



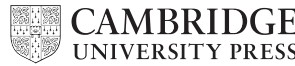

## Introduction

Despite the growing need for adolescent mental health services in the United States (US), there remain significant barriers to care, with access to treatment especially inequitable for people of

color (Alegría et al., 2008; Weersing et al., 2022; Agency for Healthcare Research and Quality, 2023). Among those seeking care for depression, people of color are considerably less likely to receive adequate care (Alegría et al., 2008). Studies have identified socio-ecological barriers that contribute to a number of disparities between adolescents of color and White adolescents. For example, stigma, caregiver attitudes, income and insurance have been found to play critical roles in access to care at the individual and structural levels (Alegria et al., 2010; Planey et al., 2019; Castro-Ramirez et al., 2021; Lu et al., 2021; Acker et al., 2023). The mental health burden and structural inequities faced by adolescents of color call for novel approaches that may bridge the services and treatment gap (Rothe et al., 2021).

One approach that is becoming increasingly studied globally to build increased access to mental health services is task-sharing (WHO, 2008). Task-sharing refers to the delivery of mental health interventions through trusted non-specialist community members, especially in underresourced settings and low- and middle-income countries (Hoeft et al., 2018; Patel et al., 2018; Lange, 2021). Importantly, a growing body of evidence is showing that scalable mental health interventions delivered by non-specialists are effective in reducing symptoms of distress and post-traumatic symptomatology across different populations and contexts at follow-up assessments (Tol et al., 2020; Zhang et al., 2020; Jordans et al., 2021; Purgato et al., 2021; Karyotaki et al., 2022; Turrini et al., 2022).

Although there is considerable work evaluating the potential benefits of task-sharing interventions globally, this approach is somewhat newer in the US. Nevertheless, burgeoning work in the US is beginning to demonstrate the feasibility and adaptability of such interventions with adults remotely (McBride et al., 2021; Pfeffer et al., 2023). Other studies have found that peer-based interventions delivered by young adults were associated with significant reductions in depression, anxiety, and loneliness, as well as higher levels of happiness, self-esteem and positive coping (Huang et al., 2018; Richard et al., 2022).

With growing interest in and evidence for task-sharing interventions as a form of scalable culturally competent care, this report introduces Early Adolescent Skills for Emotions (EASE), a recently developed World Health Organization (WHO) psychological intervention for adolescents and caregivers, to the US context (WHO and UNICEF, 2023). EASE targets psychological distress in adolescents 10–15 years of age through seven 90-minute group sessions with adolescents and three group sessions with the adolescents' caregivers (Dawson et al., 2019). Through EASE, adolescents learn how to identify their emotions, distress-related physical arousal, slow breathing as a healthy coping strategy, behavioral activation to engage in meaningful activities, and problem-solving skills (Dawson et al., 2019). Previous studies have found positive associations between emotion-focused strategies and youth mental well-being (Verzeletti et al., 2016; Fung et al., 2019), and randomized controlled trials (RCTs) have indicated the scalability and efficacy of EASE in humanitarian settings (Bryant et al., 2022a; Jordans et al., 2023; Brown et al., 2023).

The current study aimed to culturally adapt the intervention content, illustrations, and methods of implementation of EASE to center the experiences of adolescents of color in economically marginalized neighborhoods in Brooklyn, New York City (NYC). The cultural adaptation of mental health interventions recognizes the need for "systematic modification of an evidence-based treatment or intervention protocol to consider language, culture, and context in such a way that is compatible with the client's cultural

patterns, meanings, and values" (Bernal et al., 2009, p. 362). A number of studies have shown that culturally adapted interventions are more effective than nonadapted interventions for communities of color in the United States (Gonzales et al., 2016; Hall et al., 2016; van Mourik et al., 2017; Soto et al., 2018).

The adaptation process was guided by established frameworks, such as the mental health Cultural Adaptation and Contextualization for Implementation (mhCACI), which was built on Bernal et al.'s (1995) ecological validity model to outline the steps for intervention content adaptation, scalability, and implementation (Sangraula et al., 2021). Additionally, documentation of the adaptation process followed the Reporting Cultural Adaptation in Psychological Trials (RECAPT) criteria, which provides guidelines for conducting deep and surface structure adaptations (Heim et al., 2021a). Moreover, we adopted the Community-Based Participatory Research (CBPR) framework, which is a collaborative research approach that recognizes the expertise and strengths of community members, along with researchers and other stakeholders (Wallerstein and Duran, 2010). In doing so, the adaptation process was a collaborative and iterative process between researchers at The New School Center for Global Mental Health, local government at the NYC Mayor's Office for Community Mental Health, and three community-based organizations (CBOs), structured around feedback and input from adolescents and caregivers.

Through adapting the EASE intervention for underrepresented adolescents based in Brooklyn, NYC, the goal of the study was to create a version of EASE that is more accessible, relevant, and engaging for the local community and to empower mental health outcomes for underserved populations through a community-based model.

## Methods

### Participants

The NYC Mayor's Office of Community Mental Health (OCMH) identified three CBOs based in Brooklyn, NYC, to participate in the adaptation process. The CBOs primarily serve Black, Hispanic, and Latino communities in economically marginalized neighborhoods. They recruited caregivers and adolescents from their existing services to contribute to the adaptation of a new community mental health program by participating in focus group discussions (FGDs).

Caregivers who had been recipients of services from the participating CBOs were invited to attend online adaptation sessions to provide feedback on EASE caregiver sessions. For the purpose of this adaptation, a caregiver was characterized as an adult, including parents and extended family members, who routinely provided care for at least one 10- to 15-year-old adolescent. Although some of the caregivers were related to the youth participating in the adaptation process, this was not a requirement for their own participation in the adaptation.

Adolescents, primarily between the ages of 10 and 15, were recruited from existing youth programs to participate in adaptation sessions held in-person and online. Older adolescents (ages 16 and 17) were also included to gain insights into mental health challenges that may emerge over time. The adolescent participants provided feedback on the seven adolescent sessions.

We also collected feedback about the community's experiences of mental health from all adolescents and caregivers, and all participants were provided a stipend of $16 per hour for their time, as well as dinner during in-person sessions.

## Materials

The EASE intervention materials included the main publication manual, adolescent and caregiver posters, caregiver handouts, the adolescent workbook, and the storybook (WHO and UNICEF, 2023). These materials were printed out during in-person FGDs and displayed with the share screen function on Zoom/Microsoft Powerpoint during virtual FGDs.

The in-person FGDs with the adolescents also involved the use of large poster boards, sticky notes, notebooks, and colored pencils and pens for adolescents to share their thoughts with the wider group. The research team also prepared extra materials, such as emoji stickers, as alternatives for some of the EASE strategies (WHO, 2024, p. 80).

Mural (https://mural.co/), an online visual collaboration tool that allows users to create and manipulate text and images, was used in all virtual FGDs. This tool allowed adolescents and caregivers to express their thoughts online in a format mirroring the poster board and sticky notes in-person.

## Procedures

The cultural adaptation procedure was informed by the mhCACI framework (Sangraula et al., 2021) (Figure 1). The mhCACI framework is a 10-step process that guides the adaptation and implementation of evidence-based mental health interventions. It is divided into three phases. In phase I: pre-condition, investigators

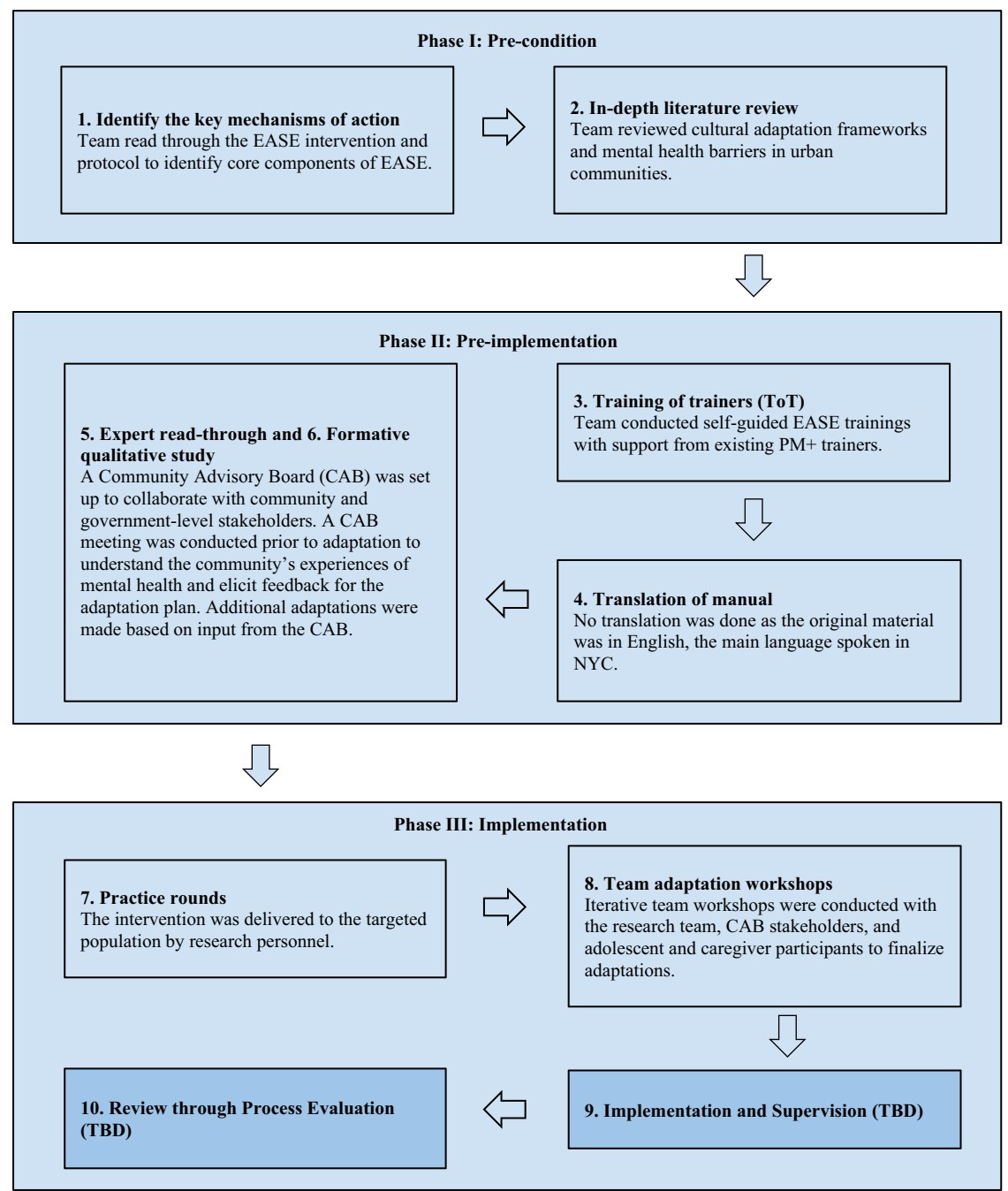

**Figure 1.** Process of adaptation guided by the mhCACI framework (Sangraula et al., 2021).

first familiarize themselves with the intervention goals and content, then they gather information on the population of interest and engagement and implementation practices to conceptualize how the intervention may be adapted. In phase II: preimplementation, investigators customize the adaptation prototype through consulting stakeholders such as intervention trainers, community members, and expert counselors. In phase III: implementation, investigators build on existing community partnerships and continue to refine the intervention as it is implemented and evaluated. As the current study focuses on documenting the process for EASE adaptation prior to pilot implementation, the last two steps of phase III: implementations (steps 9 and 10: implementation, supervision and process evaluation review) are omitted from this report.

### Phase I: Precondition

**Identify the key mechanisms of action.** The key mechanisms of action refer to the core components of an intervention that lead to improved outcomes (i.e. distress reduction) and cannot be modified drastically (Sangraula et al., 2021). The first author and study principal investigator (PI) reviewed the intervention protocol and previous trials of EASE to identify five key mechanisms of action: (1) psychoeducation and identifying emotions, (2) coping strategies for stress management, (3) behavioral activation, (4) managing problems and (5) relapse prevention (Bryant et al., 2022a; Jordans et al., 2023; Brown et al., 2023). In line with Sangraula et al. (2021), we based subsequent adaptation changes on these key mechanisms.

**In-depth literature review.** A review was conducted of cultural adaptation frameworks, adaptation of low-intensity, scalable mental health interventions and the context of adolescents' mental health globally, in the US, and in NYC. This involved searches in databases such as PubMed and PsychInfo, as well as policy briefs and annual reports from the NYC OCMH and the federal government. This step informed our understanding of the mental health barriers faced by underrepresented groups in the US.

The literature review suggested that task-sharing mental health interventions are lacking in the US and could be a way to bridge the services and treatment gap for underserved populations (Renn et al., 2023; O'Connell et al., 2024). Following the implementation of WHO task-sharing interventions in NYC CBOs, such as Problem Management Plus (PM+) for adult populations, it was determined that EASE may be a similarly viable and valuable resource catered to the city's adolescents.

### Phase II: Preimplementation

**Training of trainers (ToT).** The first and second authors were trained as EASE trainers in the current study (see Supplemental File 1 for their demographic characteristics). The ToT process did not involve formal training conducted by EASE trainers from previous study sites, following Sangraula et al.'s (2021) protocol; instead, the first and second authors conducted self-guided EASE training. The self-guided training involved reviewing the EASE manual in detail and discussing techniques to strengthen the key mechanisms of action. This self-guided training was approved by the WHO with ad hoc WHO technical support as needed due to the research center's extensive experience with adapting and implementing PM+, a similar transdiagnostic intervention (Dawson et al., 2019; Bryant et al., 2022b; Pfeffer et al., 2023; Kohrt et al., 2025). The study PI, who is an experienced PM+ trainer, supported the EASE ToT process by helping trainees understand strategies shared by both interventions.

Training sessions were conducted prior to each corresponding FGD session so that iterative adaptation changes to the ways in which EASE activities were carried out could be made based on feedback from previous FGDs. Such changes focused on enhancing participant engagement or amplifying key mechanisms of action.

**Translation of manual.** Although many languages are spoken in NYC, this adaptation focused on English speakers, with 75% of the city proficient in the language (NYC Department of City Planning, 2025). Hence, we did not conduct any translations to the intervention material, which is in English. However the original intervention is in British English and contains language that does not align with the colloquial language of adolescents and caregivers in NYC. Thus, the research personnel discussed ways to provide explanations during FGDs using culturally relevant terminology that resonates with our target population.

**Expert read-through and formative qualitative study.** Steps 5 (expert read-through) and 6 (formative qualitative study) aim to gain information on the applicability of the intervention materials from experienced persons in the program site and understand the targeted community's awareness of mental health, preexisting community mental health resources and barriers faced by the community (Sangraula et al., 2021).

To this end, we set up a Community Advisory Board (CAB) per CBPR guidelines (Newman et al., 2011) to collaborate with community and government-level stakeholders. The committee included the study's PI, members of The New School's Center for Global Mental Health (see Supplementary File 1), members of OCMH, and key staff members from the three CBOs in Brooklyn. The research team met with the CAB to plan for the FGDs. The CAB meeting involved CBO staff members ($n = 5$), an FGD caregiver participant ($n = 1$), OCMH members ($n = 2$), and members of the research team ($n = 2$). The first author took notes on the community stakeholders' feedback. These notes were relayed back to the research team to devise the next steps of the adaptation process. Key observations and the subsequent adaptation changes are summarized later.

The CBO staff expressed that opportunities for adolescents to lead discussions would encourage active participation. As a result, FGDs were designed in blocks of intervention content and immediate feedback discussion on each section of the storybook, workbook, and strategy covered.

Adolescent FGDs were hosted in person and virtually to examine the feasibility of delivering EASE remotely. As suggested by the CAB, the first FGD session was designed as a virtual orientation with all participants to build rapport and introduce the study. Afterwards, the adolescents and caregivers were split into separate FGDs. The caregivers attended three additional virtual FGDs (1.5 hours each), whereas adolescents participated in seven additional FGDs, consisting of three in-person FGDs located at each of the three CBOs (3 hours each) and four virtual FGDs on Zoom (1.5 hours each). The all-virtual approach to caregiver FGDs was planned in accordance with the feasibility and preference of adults for virtual PM+ training (McBride et al., 2021), as well as CAB input that virtual sessions best complement busy caregiver schedules. All virtual sessions involved a technical facilitator to troubleshoot technological difficulties.

Based on input from CAB stakeholders, the adolescent FGDs were further divided into two sub-groups with younger and older adolescents (1) to encourage discussion and initiative-taking from the adolescents in smaller settings, (2) to keep each adolescent

group size from 8 to 12 participants per WHO guidelines, and (3) to explore the suitability of EASE for adolescents in varied developmental stages.

### Phase III: Implementation

Practice rounds. The FGDs were composed primarily of practice rounds, which refer to intervention delivery to the targeted population to obtain first-hand feedback about adaptation (Sangraula et al., 2021). In each FGD, two members of the research team served as facilitators leading intervention delivery and guiding group discussions, and two members served as notetakers. CBO staff also participated in the adolescent FGDs to help guide discussion. Following each practice round delivery of a core EASE component, participants engaged in extensive discussion focused on providing feedback. Cognitive interviews about the adolescents' overall experience with EASE were also conducted during the final in-person FGD. The questions from the cognitive interviews can be found in Box 1 of Supplementary File 2.

Team adaptation workshops. Community sharebacks in June and October were conducted with the research team, CAB stakeholders, and adolescent and caregiver participants to review all intervention adaptations following the completion of practice rounds. In both sharebacks, the research team presented an overview of adaptation changes to date and obtained additional feedback from all stakeholders. The recommendations were then reviewed for final adjustments.

### Data analysis

Data analysis was conducted on the four main components of the RECAPT criteria: (1) cultural concepts of distress; (2) community needs, stigma and context; (3) treatment components and (4) treatment delivery (Heim et al., 2021a). When mapping out FGD themes, we combined the deductive approach of the RECAPT criteria with an inductive approach based on participant feedback and research team observations.

As adolescents and caregivers were divided into FGD groups of six to seven participants, data from FGDs were manually coded by the research team. The FGD notetaker jotted down participants' feedback and observations during the FGDs, and these notes were reviewed by the research team to identify themes. In addition, we collected participants' immediate feedback via sticky notes during in-person sessions and via Mural during virtual sessions.

Data were extracted from the FGD notes, including responses from cognitive interviews. The team also conducted post-FGD debriefings to discuss the observations and feedback from all FGDs, with the goal of reaching a consensus on the key themes of observation and feedback from each FGD. Key points were summarized into themes based on the RECAPT criteria by the first two authors.

### Results

### Participant characteristics

As participants were recruited directly from the community programs of the partner CBOs, which serve Black, Hispanic, and Latino communities in economically marginalized neighborhoods in Brooklyn, NYC, the participant group reflects the communities and cultures that EASE aims to serve. The average age of the adolescents ($n = 18$) was 12.9, reflecting the EASE intervention's

targeted range. The group was relatively balanced in gender and was largely composed of adolescents who identified as Black or African American (72%), with the remaining identifying as Hispanic or Latino (22%) and American Indian (6%), as reported by participants in a questionnaire based on the racial and ethnic identity categories used by the US Census Bureau (2024). A number of adolescents were second generation immigrants (42%) or first generation immigrants (7%), and bilingual or multilingual (21%). The majority were in families receiving Medicaid (income-based health support) or SNAP (nutrition support) benefits (60%), and some adolescents had experienced homelessness (14%) (see Table 1). Twelve caregivers from the three partner CBOs participated in the study. Eight participants were caregivers of the adolescents enrolled in the study. Unfortunately, the demographic characteristics of the caregivers were not collected in the current study as caregivers were enrolled a couple weeks after adolescents and thus did not have a chance to complete demographic questionnaires.

### Adaptation

This section highlights selected adaptation themes documented from the FGDs reported following the RECAPT criteria (Heim et al., 2021a), and can be seen in full in Supplemental File 1. Initially conceived to standardize and document the cultural adaptation process of mental health RCTs for refugees in Germany and reviewed by experts in cultural adaptation literature, the RECAPT consists of 11 criteria grouped under four stages (Heim et al., 2021a).

Stage A: A setup included documenting key characteristics about the target population and the researchers' backgrounds (see Supplemental File 1).

**Table 1.** Sociodemographic characteristics of adolescents

|  | M (N, %) | SD |
|---|---|---|
| Age | 12.89 | 1.82 |
| Gender |  |  |
| Male | (10, 55.56%) |  |
| Female | (8, 44.44%) |  |
| Racial identity |  |  |
| Black or African American | (13, 72.22%) |  |
| Hispanic or Latino | (4, 22.22%) |  |
| American Indian | (1, 5.56%) |  |
| Experience |  |  |
| Bilingual/multilingual | (3, 21.43%) |  |
| First-generation immigrant | (1, 7.14%) |  |
| Second-generation immigrant | (5, 41.67%) |  |
| Experience in foster care | (1, 7.14%) |  |
| Have been homeless | (2, 14.29%) |  |
| Juvenile justice system experience | (0, 0.00%) |  |
| IEP in school | (2, 15.38%) |  |
| Medicaid (income-based health support) or SNAP (nutrition assistance) benefits | (6, 60.00%) |  |

*Note*: Not all participants responded to all of the questions about their lived experiences, hence the percentages reported only reflect the proportion of participants who responded.

Stage B: Formative research outlined the methods for the cultural adaptation process and reported on the target population's mental health context and experiences, including (1) cultural concepts of distress; (2) community needs, stigma and context; (3) treatment components; and (4) treatment delivery.

Stage C: Intervention adaptation reported changes made to specific treatment elements based on the key mechanisms of action as well as surface adaptations aimed to improve intervention acceptability by our target population.

Stage D: Measuring outcomes and implementation includes criteria on reporting the questionnaires, clinical interviews and processes used during implementation (Heim et al., 2021a). These criteria are omitted in the current report as the study focuses on the adaptation of EASE prior to pilot implementation.

### Cultural concepts of distress

Cultural concepts of distress refer to the ways in which the target population perceives and experiences distress. This category is subdivided into idioms of distress based on different symptoms (e.g. emotional, behavioral), explanations of distress, and beliefs about the distress (Heim et al., 2021c; Aeschlimann et al., 2024). The current report focuses on the ways in which distress is conceptualized by our surveyed sample due to its influence on the subsequent adaptation changes.

Notable cultural explanations of distress included a lack of understanding of distressing emotions, lack of knowledge of how to respond to emotions, and the need to hide emotions. During the discussion about identifying feelings in the second FGD, adolescents shared their challenges with explicitly labeling their emotions. Adaptations were made based on these results to add a "Feelings Wheel," which introduced a richer emotional vocabulary necessary for the identification and communication of nuanced feelings containing over 50 emotion words. This complemented and extended the original intervention's "Feelings Chart" of 10 basic emotions as part of Adolescent Activity 1.5 (Understanding my feelings).

In the second FGD, adolescents cited that they sometimes felt controlled by their emotions in response to events that made them sad or angry. While the original activity focused on brainstorming problems and potential solutions, we added the concept and activity of a "Circle of Control" to Adolescent Activity 6.4 (Managing my problems with a new problem), which encourages adolescents to reflect on which aspects of a stressful situation are within their control and, thus, where to focus their energies in managing their emotions and efforts to change their external environments. Moreover, adolescents shared that they often conceal their emotions to appease societal expectations, especially among young men. In relation to this finding, some of the participants indicated that the "Feelings Pot" in Adolescent Activity 1.7 (Identifying personal feelings) may reinforce the idea of suppressing instead of learning to express one's emotions. For example, when completing the activity, an 11-year-old said: "Why are we putting emotions into a pot? That's just like bottling them up, but we're supposed to learn how to express feelings."[1] Thus, the Feelings Pot in the original intervention was changed to a Feelings Canvas to encourage emotional expression (see Figure 2). This modification was met with great enthusiasm by adolescents during the community shareback in June 2024.

---

[1]Other notable quotes from cognitive interviews can be found in Box 2 of Supplementary File 2.

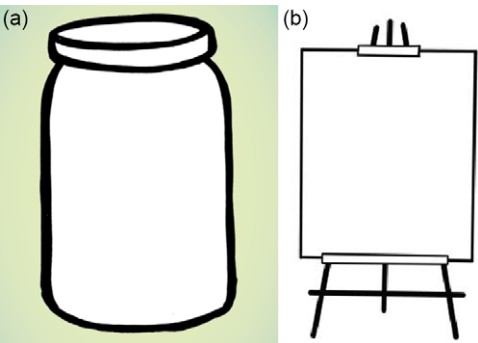

**Figure 2.** Adaptation of Feelings Pot to Feelings Canvas in Adolescent Activity 1.7 (Identifying personal feelings). (A) Feelings Pot in the original manual. (B) Feelings Canvas, which is the adapted version of the Feelings Pot, in the current manual. These images are reproduced and adapted from Early Adolescent Skills for Emotions. Geneva: World Health Organization and the United Nations Children's Fund (UNICEF), 2023. License: CC BY-NC-SA 3.0 IGO. WHO is not responsible for the content or accuracy of this translation/adaptation.

### Community needs, stigma and context

Community needs, stigma and context refer to the setting-specific information that influences mental wellbeing (Heim et al., 2021c). This includes factors such as the community's understanding of mental health based on sociocultural determinants, openness to treatment, and access to mental health resources. This category is divided into two subcategories: attitudes toward mental health, which reflect how the community considers mental health aside from their perception of distress, and specific needs and other contextual information, which outlines social determinants that shape the current adaptation.

**Attitudes toward mental health.** Adolescents and caregivers' attitudes toward mental health shared in FGDs included advocacy for inclusive and safe spaces when sharing mental health struggles; a desire for deeper knowledge-based explanations of psychological benefits; and building purpose, meaning, connection and identity.

Both adolescent and caregiver participants saw the promotion of inclusivity and sharing in safe spaces as important tenets of mental health. To that end, the original intervention's "Body Map" poster of one male outline was expanded into five additional versions of outlined bodies that reflect different genders and body shapes (Adolescent Activity 2.3, Feelings and My Body). Associations of shame and physical punishment as parental discipline were also removed from the caregiver story (Caregiver Activity 2.5, Alternatives to harsh punishment). These adaptations aimed to reduce potential stigma and judgment such that participants could feel safer expressing themselves.

Moreover, the adolescents' curiosity about the mind–body connection and the physiological basis of the coping strategies they were learning led to the addition of a Slow Breathing infographic in Adolescent Activity 2.4 (Calming my body) (see Figure 3). This infographic provides in-depth information about how the Slow Breathing strategy is connected with the nervous system. Variations to the strategy were also included to better connect it with the adolescents' interest in somatic experiences. These adaptations aimed to promote engagement and relevance of the Slow Breathing activity.

Throughout the FGDs, adolescents expressed a strong desire to develop self-understanding and navigate purpose and meaning, particularly in relation to seeing how their personal identities might relate to the communities around them. These themes were mostly

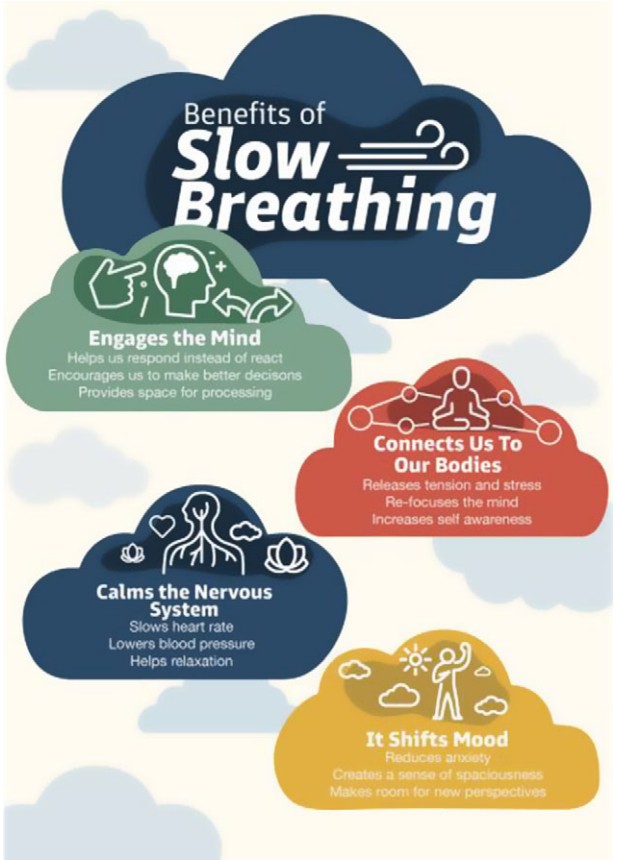

**Figure 3.** Slow Breathing infographic in Adolescent Activity 2.4 (Calming my body). The infographic provides in-depth information about how the Slow Breathing strategy is connected with the nervous system.

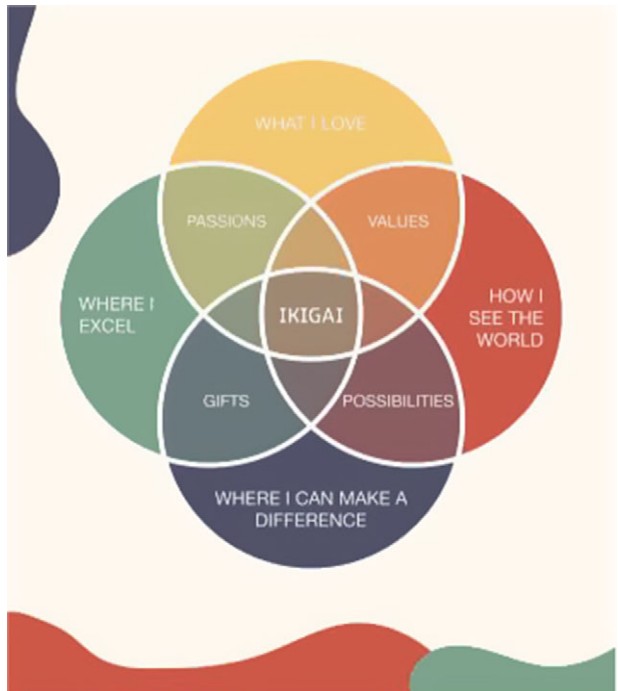

**Figure 4.** Purpose Diagram in Adolescent Activities 3.4 and 4.3 (Changing my actions); 6.5 (Completing the Purpose Diagram); and 7.3 (Brighter Futures). Through several Purpose Diagram activities, adolescents complete a full Purpose Diagram.

found in the fourth, fifth, and sixth FGDs, which covered behavioral activation and problem management strategies. For instance, when completing Adolescent Activity 3.4 (Changing my actions) and 5.4 (Managing my problems), adolescents questioned how they could identify a goal to work toward or how they could address a community-level concern based on their personal interests, strengths, and weaknesses. Thus, new activities were added to better connect EASE strategies to the adolescents' personal lives and communities. For example, the concept and activity for a "Virtuous Cycle" was added to Adolescent Activity 4.3 (Changing my actions) to complement the original "Vicious Cycle." The new activity encourages adolescents to explore how a self-reinforcing loop of fulfilling activities can motivate them. Similarly, the concept of the "Growth Mindset" and "Growth Zone" as opposed to the "Fixed Mindset" and "Comfort Zone" was included in Adolescent Activity 5.3 (Understanding common problems). This activity aims to help adolescents reframe challenges as learning opportunities. Furthermore, "Purpose Diagram" activities were included in Adolescent Activities 3.4 and 4.3 (Changing my actions) (see Figure 4). These initial activities prompt adolescents to consider what things they love doing, what things they are good at, and the intersection of these domains to identify their "passion." The final Purpose Diagram activities are introduced in Adolescent Activities 6.5 (Completing the Purpose Diagram) and 7.3 (Brighter Futures) to explore "how I see the world," "how I can make a difference," and the intersecting components of "values," "gifts" and "possibilities." The Purpose Diagram activities aimed to guide adolescents in reflecting on how different aspects of their lives are interconnected and how they can form meaningful goals.

**Specific needs and other relevant contextual information.** Gender norms emerged as an important point of discussion during the caregiver FGDs. The caregivers felt that the original storyline emphasized the female caregiver's homemaking responsibilities in comparison to the male caregiver but failed to discuss the emotional impact of this traditional gender norm equitably. To this end, a greater male caregiver presence was incorporated in the storyline, and we also included a discussion about caregiver responsibilities that could be influenced by gender roles in Caregiver Activity 3.4 (Gender roles in caregiving). We aimed to acknowledge the contributions of all caregivers and prompt discussion about ways to create a more supportive caregiving environment.

Furthermore, caregivers and adolescents expressed how digital technology could affect youth mental health positively and negatively throughout the FGDs. Thus, in the storybook adaptation, we portrayed the benefits of digital technology through the main character's use of his phone to learn skateboarding skills and to connect with his friends. In contrast, the harms of digital technology are represented by cyberbullying, which the main character learned to address with the EASE skill of problem-solving.

In addition to cyberbullying, adolescents reported facing challenges from the economic pressures of NYC's high living costs and peer pressure. We incorporated these findings in the adapted storybook to make it more relevant. For example, we included a scene of the main character struggling with pocket money when his friends wanted to go to a bodega together (Storybook text 19).

Regarding group dynamics, suggestions from the final FGD with adolescents emphasized the importance of bonding with facilitators and building trust among the group to encourage more honest and open expression. The older adolescents particularly appreciated

how the CBO staff helper in their group shared personal experiences. This openness made the adolescents feel more comfortable sharing their own thoughts and feelings. Appropriate and genuine self-disclosure by EASE facilitators should be encouraged during training sessions. Although activities to build rapport within EASE groups exist in the original manual (e.g. Welcome activity at the beginning of all adolescent sessions), they were considered "too young" for the 10 to 15 year old age group in NYC. Thus, we replaced icebreaker suggestions like singing a song in the group with the use of multimedia, such as presenting pop culture video clips related to EASE topics for discussion. Similarly, roleplays were found to create awkward dynamics among adolescent groups, and adolescents were not actively engaged in them. For example, a 15-year-old commented that the roleplay activities were "*weird, childish and not fun*"[2] during the final FGD. As a result, we de-emphasized the use of roleplays throughout the adolescent sessions. For instance, the roleplay activity for "linking feelings and behaviors" in Adolescent Activity 4.3 (Changing my actions) was replaced by a visualization activity that served the same purpose: encouraging adolescents to be more emotionally aligned with their goal.

### Treatment components

Treatment components refer to the ways in which the intervention is oriented and structured, taking into account the community's understanding of distress, lived experiences, and access to mental health resources. Echoing Aeschlimann et al. (2024), the current study considers how the intervention goals are framed to address community mental health concerns. We found that it is crucial to clarify the framing of treatment goals when considering how to modify EASE treatment components. Two treatment goals were identified in the study: (1) developing social and emotional learning (SEL) and (2) community-building.

**Framing treatment goals.** As the current adaptation aims to create a version of EASE catered to urban adolescents and caregivers in NYC, we aimed to align the adaptation to programming by the Department of Youth & Community Development (DYCD) in NYC, which integrates SEL competencies identified by the Collaborative for Academic, Social and Emotional Learning (CASEL, 2015). CASEL addresses the five competencies of self-awareness, self-management, social awareness, relationship skills, and responsible decision-making (CASEL, 2015). We found that the CASEL competencies of self-awareness, self-management, and responsible decision-making seemed to be most salient in the original EASE activities, as evidenced by the adolescents' feedback in the second FGD, which covered the EASE theme of identifying and understanding feelings, and the fifth and sixth FGDs, which covered the EASE theme of problem management. For instance, Adolescent Activity 1.7 (Identifying personal feelings) encouraged adolescents to use colors as a way of expressing emotions through the Feelings Pot. This activity prompts adolescents to better articulate their feelings, thus building their emotional regulation skills and competency for self-management: managing one's emotions, thoughts and behaviors (CASEL, 2015). Moreover, Adolescent Activities 5.4 (Managing my problems) and 5.5 (Applying managing my problems), which introduced the Stop, Think, Go strategy to guide adolescents in

brainstorming solutions for their problems, is related to the competency of responsible decision-making, which includes the ability of developing one's critical thinking skills and making constructive choices (CASEL, 2015). However, these two competencies, as well as the competencies of self-awareness, social awareness, and relationship skills, could be more greatly emphasized in the current adaptation to better align EASE with DYCD programming.

The current adaptation's Purpose Diagram activities (Adolescent Activities 3.4 and 4.3, Changing my actions; Adolescent Activity 6.5, Completing the Purpose Diagram; Adolescent Activity 7.3, Brighter Futures) target all of the CASEL competencies of SEL. Through activities that encourage adolescents to reflect on what they love and what they are good at, adolescents can improve their self-awareness through reflecting on their identities and strengths. Furthermore, through discussions about how they see the world and how they can make a difference, adolescents are guided to consider how they may contribute to their communities, thereby developing the competencies of social awareness, relationship skills, and responsible decision-making.

These new SEL-related additions to the program are supported by previous research that emphasizes the importance of SEL for adolescents in various contexts (Cherewick et al., 2021; Marsay et al., 2021; Gimbert et al., 2023; Maloney et al., 2024; Martinez and Gomez, 2024). Given existing literature highlighting the significance of mentalizing and emotional regulation to self-development (Pfeifer and Peake, 2012) and how SEL practices may benefit the development of ethnic-racial identity (Rivas-Drake et al., 2020), the Purpose Diagram activities aimed to further promote these areas of growth.

Community-building was identified as a second major treatment goal. This theme emerged throughout all the adolescent FGDs, and it is related to the previous finding on group dynamics. However, community-building goes beyond simply fostering rapport among adolescents and facilitators to encourage more open and honest sharing. We found that the treatment goals of SEL and community-building may complement each other. Notably, while reviewing the Slow Breathing home practice during the final FGD, a 14-year-old mentioned enjoying the activity with their friends, saying "Yeah we breathed together, they calmed me down, they're real friends. Real friends breathe together!"[3]

This finding was reinforced during team adaptation workshops, where CBO staff members emphasized that the original intervention's group discussions and activities could create a safe space for adolescents to learn collaboratively. The CBO staff members also expressed that developing adolescents' socioemotional skills could potentially improve community-building efforts by equipping them with the ability to establish and maintain healthy relationships. These findings point towards the importance of prioritizing community-building as a central goal of EASE. To this end, we added an EASE "yearbook" activity and expanded the graduation ceremony in Adolescent Activities 7.3 (Brighter futures) and 7.4 (Ending the intervention). Additionally, we are exploring the possibility of optional post-EASE drop-in sessions that can be integrated into existing community programs, such that adolescents can come together following the end of the official program to build relationships and continue practicing EASE skills. Through celebrating adolescents' accomplishments in the EASE program, encouraging them to write and draw messages to future participants, and creating spaces for adolescents to continue learning from

---

[2]Other notable quotes from cognitive interviews can be found in Box 2 of Supplementary File 2.

[3]Other notable quotes from cognitive interviews can be found in Box 2 of Supplementary File 2.

each other, we aim to promote community connectedness and program sustainability.

The importance of community-building in supporting adolescents' socioemotional learning and growth aligns with previous studies indicating that adolescents responded to interviews and therapy more productively when interviewers and therapists offered self-disclosure or engaged in deliberate rapport-building (Brown et al., 2014; Dianiska et al., 2021, 2024). In forming trusted connections with other participants and the EASE facilitators and CBO staff helpers, participants were more likely to share their thoughts and feelings honestly without fear of judgment. This is supported by past studies indicating how community-based approaches could enhance SEL programming for adolescents, caregivers, and educators (McKay-Jackson, 2014; Anziom et al., 2021; Paik et al., 2024; Speidel et al., 2024).

### Treatment delivery

Treatment delivery refers to how the intervention is implemented, including the format, method of communication, and the ways in which the key mechanisms of the intervention are portrayed to the community (Heim et al., 2021b, 2021c; Aeschlimann et al., 2024). This category is subdivided into delivery format and surface adaptations, and the following sections outline findings and modifications made to improve the acceptability and engagement of EASE within the target community. Results regarding treatment delivery highlight issues and recommendations related to the delivery format, such as barriers to and advantages of use, promotion of use and key takeaways from surface adaptations.

**Delivery format.** Through the caregiver FGDs, we found that virtual meetings increased accessibility and convenience. The caregivers expressed preferences for virtual delivery, citing benefits of flexibility for their busy schedules. However, for adolescents, we observed more engagement, better focus and information absorption in in-person FGDs than in virtual FGDs; thus, we determined that in-person delivery is more suitable for adolescents. This result is in line with previous findings indicating that adolescents value forming connections with facilitators and other EASE group members, which helps them feel more comfortable sharing their thoughts and feelings, and aligns with the treatment goal of community-building.

In light of the community-building treatment goal, results indicated that the delivery of EASE could be integrated into existing adolescent groups within community organizations or after-school programs. The established relationships among members and trusted environments in these settings offer safe spaces that can support adolescents in honest self-expression and promote more effective SEL. These methods of delivery will inform the pilot implementation of EASE in the next phase of the study, where we aim to collaborate with DYCD to bring this adapted version of EASE to CBOs in NYC.

**Surface adaptations.** Throughout the FGDs, adolescents raised the critical point about not being able to identify with the characters and storyline in the EASE storybook. After reading storybook texts that involved the storybook main character encountering colorful, talking birds and engaging in birdwatching as a hobby, a 15-year-old said "In NYC, if you're a kid, you're probably walking to school alone, unless you take the bus or train to go to school in another area, but you can't talk to birds in the subway or through the window. Seeing birds while walking on the sidewalks to school

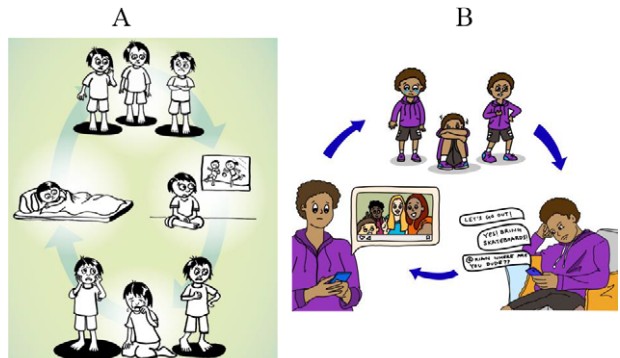

**Figure 5.** Adaptation of the Vicious Cycle Poster in Adolescent Activity 3.3 (Feelings and actions). (A) Vicious Cycle Poster in the original version. (B) Adapted version of the Vicious Cycle Poster in the current manual. These images are reproduced and adapted from Early Adolescent Skills for Emotions. Geneva: World Health Organization and the United Nations Children's Fund (UNICEF), 2023. License: CC BY-NC-SA 3.0 IGO. WHO is not responsible for the content or accuracy of this translation/adaptation.

makes more sense, I guess, but it's not a hobby."[4] And a 13-year-old said, "Why is he still with the bird? Are the birds in his head?"[5] as the storybook main character continued developing his relationship with the birds. To make the main character more relatable to the adolescents, we modified the main character's hobby to skateboarding, something that the adolescents expressed was a fun, enjoyable, and trendy activity. We removed the motif of the colorful talking birds and replaced it with an artist from a community center using different colors on his canvas to express emotions. This change aligns with the goal of Adolescent Activity 1.7 (Identifying personal feelings), which intended to use the birds to introduce adolescents to the concept of using colors to identify their feelings. Moreover, through incorporating themes such as gender norms, use of technology, economic pressure, and peer pressure in both the adolescent and caregiver storylines, we aimed to better reflect the challenges faced by adolescents and caregivers from the target community, and the storybook, poster and workbook illustrations were modified to be more representative of people in NYC (see Figure 5). These changes were highly endorsed by adolescents and caregivers during the community shareback in October 2024, with many participants sharing that the revised storyline and characters more accurately reflected their day-to-day experiences in NYC. Adolescents also expressed greater interest in this version of EASE because they could better identify with the main character's journey of self-emotional exploration.

Finally, throughout the FGDs, the adolescents often proposed changes to phrases, idioms, and word use. In particular, both the younger and older adolescent groups expressed that the language used in the storybook seemed too formal. Many issues surrounding language were also related to the content of the adolescent storybook. For example, regarding storybook text 2, a 12-year-old said: "Who says 'can we sing together'? It doesn't sound New York at all. I'd ask if his sister would want to watch TV or play a game."[6] Thus, local idioms and colloquial language were incorporated in the final text to make it more acceptable to the adolescents.

---

[4]Other notable quotes from cognitive interviews can be found in Box 2 of Supplementary File 2.

[5]Other notable quotes from cognitive interviews can be found in Box 2 of Supplementary File 2.

[6]Other notable quotes from cognitive interviews can be found in Box 2 of Supplementary File 2.

## Activity ratings

In addition to collecting feedback based on the RECAPT criteria, we asked the adolescents to rank key elements (activities and posters) of the EASE intervention by (1) how helpful and (2) how engaging they found them out of 5. The average scores for these individual ratings, as well as the average scores across both ratings, can be found in Table 2.

Notably, the Body Map activity (Adolescent activity 2.3), the Stop, Think, Go activity (Adolescent activity 5.4), the Staircase activity (Adolescent activity 3.4) and the Solutions Tree activity (Adolescent activity 5.5) were in the top five highest rated elements for both how helpful and how engaging they were for the adolescents.

**Table 2.** Average element ratings for core EASE elements

| Element | Overall score | Helpfulness score | Engagement score |
|---|---|---|---|
| Body Map (Adolescent activity 2.3)[1] | 4.45 | 4.80 | 4.17 |
| Stop, Think, Go (Adolescent activity 5.4)[2] | 4.36 | 4.30 | 4.40 |
| Staircase (Adolescent activity 3.4)[3] | 4.27 | 4.31 | 3.90 |
| Coping Strategies (Adolescent activity 2.4)[4] | 4.25 | 4.25 | * |
| Solutions Tree (Adolescent activity 5.5)[5] | 4.20 | 4.40 | 4.00 |
| Vicious Cycle (Adolescent activity 3.3)[6] | 3.94 | 4.25 | 3.63 |
| Maze Poster activity (Adolescent activity 5.3)[7] | 3.89 | 3.83 | 4.00 |
| Slow Breathing (Adolescent activity 2.4)[8] | 3.75 | 4.00 | 2.00 |
| Feelings Pot (Adolescent activity 1.7)[9] | 3.73 | 3.94 | 3.40 |
| Feelings Chart (Adolescent activity 1.5)[10] | 3.63 | 3.54 | 3.71 |
| Sadness Poster (Adolescent activity 1.6)[11] | 2.71 | 2.80 | 2.64 |

*Note:* The activities are presented in descending order of their overall score. Not all adolescents rated all of the activities, hence the scores reported do not reflect the averages for all participants.

*No engagement score is reported because the adolescents did not rate it.

[1] Through drawing physical sensations associated with emotional experiences on body outlines, adolescents explore how their bodies may be affected by problems and feelings.
[2] Through going through steps for problem management, adolescents learn to define a practical problem, explore possible solutions, choose the most helpful option, and plan to solve it.
[3] Through filling in a plan for carrying out an enjoyable or meaningful activity, adolescents learn to break down their goals into small, manageable steps.
[4] Through discussing coping strategies, adolescents learn to distinguish between helpful and unhelpful approaches.
[5] Through writing potential solutions on the branches of a tree outline, adolescents practice brainstorming different ways to solve a problem.
[6] Through discussing how the storybook main character is affected by a vicious cycle, adolescents learn how intense emotions can make one feel worse.
[7] Through working together to find their way through a maze, adolescents learn the importance of trying out different options to solve a problem.
[8] Through discussing the benefits of slow breathing and practicing it together, adolescents learn a skill to calm their bodies.
[9] Through drawing feelings in the feelings pot, adolescents learn to express their emotions using creative means.
[10] Through identifying the different feelings portrayed by characters in the feelings chart, adolescents learn to recognize and label emotions.
[11] Through identifying features of a character portraying sadness, adolescents learn how feelings can be manifested in different ways.

This feedback helped guide the adaptation of activities and illustrations, as some elements were found to be helpful but less engaging, suggesting that the elements could be made more relevant and appealing to NYC adolescents. For instance, the Slow Breathing strategy (Adolescent activity 2.4) was rated as fairly helpful (4.00) but less engaging as presented in the original manual (2.00). This finding echoed observations and quotations from the adolescents, as outlined in the previous section, indicating that the adolescents appreciated the strategy's potential benefits but did not find it interesting. As a result, we added graphics and workshopped additional variations to the activity to make it more engaging.

## Discussion

This paper presents findings of the community-based adaptation of EASE for underserved youth and caregivers in NYC. Through a series of FGDs with stakeholders such as youth, caregivers and CBO staff members, we obtained a range of valuable feedback regarding the areas of (1) cultural concepts of distress, (2) community needs, stigma and context, (3) treatment components and (4) treatment delivery. The feedback reflected the need for mental health resources and equitable access to mental health treatment for underrepresented groups in the US (Garland et al., 2005; Alegría et al., 2008; Martin et al., 2021; Rothe et al., 2021; Chen et al., 2022; Fan et al., 2022; Rodgers et al., 2022; Weersing et al., 2022).

We found that adaptation frameworks were essential for guiding our understanding of the intervention's core elements, the target population and context, and the changes made. In particular, the RECAPT criteria enabled us to comprehensively assess community needs and treatment approaches (Heim et al., 2021a). Based on the RECAPT, we made deep adaptations that aligned the intervention's components with SEL goals (CASEL, 2015) and emphasized its orientation to community-building. We also made surface adaptations to improve EASE's acceptability, relevance, and engagement, such as altering the depiction of characters, incorporating community-specific day-to-day experiences and challenges, and using local idioms. The mhCACI framework, on the other hand, helped prepare us for future implementation and scaling. Its emphasis on adapting interventions within community settings and iterative processes has encouraged our ongoing collaboration with the OCMH, the three partnering CBOs, and participating adolescents and caregivers (Sangraula et al., 2021). Echoing existing literature, we found that the use of systematic frameworks streamlined the adaptation process and facilitated the integration of evidence-based knowledge, cultural contexts, and close partnership with community stakeholders (Wang et al., 2018; Leung et al., 2024; Fernández et al., 2025).

Results preliminarily suggest that the process of culturally adapting EASE was an important first step toward its implementation. Although it is still to be determined, the steps taken to adapt EASE in collaboration with various stakeholders from the community may result in greater uptake and engagement, as these steps are in line with both international recommendations (Inter-Agency Standing Committee, 2007, as well as frameworks and proposals underscoring the importance of adaptation prior to implementation (Bennouna et al., 2019; Perera et al., 2020). Importantly, a burgeoning body of research has found that culturally and contextually adapted interventions are associated with factors such as greater fit, acceptability, relevance, and fidelity (Bernal et al., 1995; Ferrer-Wreder et al., 2012; Fernández et al., 2025).

Arguments in support of the potential benefits of adaptation towards implementation were also supported by feedback from

stakeholders in our study. Throughout the FGDs, adolescents and caregivers indicated that the intervention strategies had to be relevant to community needs, the characters and scenes needed to be representative of their day-to-day experiences and the challenges, and coping strategies recommended in the EASE curriculum had to feel realistic in order for implementation to be successful. All stakeholders also expressed great enthusiasm during the community sharebacks, where we shared preliminary changes made to the intervention based on participants' feedback.

Interestingly, another potential reason that this process may lead to greater implementation success may come from directly engaging adolescents in FGDs. Rather than simply relying on adults to guide potential implementation strategies, we prioritized youth perspectives to ensure that the adaptation would be more aligned with the actual realities of the young people it aims to serve. The inclusion of youth in the adaptation of interventions may be an important factor in implementation success, and it will be helpful to see how youth participation contributes to outcomes in various contexts (e.g. Galbraith et al., 2023; Freeman et al., 2024). Moreover, the adaptation processes may allow for greater implementation success as the FGDs were likely viewed as opportunities for community-building. Framing adaptation as a form of community-building builds on prior research that suggests the significance of leveraging relational factors to engage mental health non-specialists in co-designing and implementing community-based programs (Castro et al., 2004; Marsiglia and Booth, 2015; Simpson et al., 2022; Norman et al., 2024).

Additionally, the results are supported by the qualitative rigor of data collection and analysis. For instance, purposive sampling was conducted to identify adolescents of color in economically marginalized neighborhoods in Brooklyn, NYC, through collaboration with local CBOs. This procedure ensured that the participants and contexts surveyed were most appropriate for meeting the research goals (Johnson et al., 2020).

Furthermore, the study aimed to maintain a high degree of trustworthiness in its methods. Trustworthiness ensures result reliability through the credibility, transferability, dependability, and confirmability of qualitative findings (Stahl and King, 2020; Ahmed, 2024). The current study established credibility via data triangulation, as we used several sources of information, such as the researchers' observational feedback and interview notes, to identify patterns (Johnson et al., 2020; Stahl and King, 2020). In addition, the study involved prolonged engagement, such that researchers gained familiarity with the participants and their cultures and contexts over the course of 2 months. This strategy enhanced result credibility and supported researcher reflexivity (Johnson et al., 2020; Stahl and King, 2020). By discussing individual backgrounds and experiences (see Supplemental File 1), the team was better able to recognize potential research biases and report findings that more accurately represent participants' perspectives. To establish confirmability, we conducted member checking and peer debriefing, which allowed participants and colleagues at the NYC Hall to review our work (Busetto et al., 2020; Johnson et al., 2020; Stahl and King, 2020; Ahmed, 2024). Regarding transferability, Supplemental File 1 provides thick descriptions on the research procedures, contextual information, and decisions made based on the RECAPT criteria to provide insights into the applicability and relevance of our findings to other contexts (Stahl and King, 2020; Ahmed, 2024). Our documentation of the adaptation process also ensures traceability for

potential study replication, thus reaching the criterion of dependability (Johnson et al., 2020; Ahmed, 2024).

### Limitations and future directions

As the current study is limited to reporting the adaptation process, future studies are needed to assess the adapted intervention within an implementation context to determine the significance of cultural adaptation towards effective implementation. As outlined above, we omitted steps 9 and 10 of the mhCACI framework in our methods; hence, the current study only provides a preliminary indication of how cultural adaptation could lead to more effective implementation. We decided to publish this report ahead of implementation for a more detailed documentation of the adaptation process, addressing the current gap in adaptation reports and contributing to the literature by illustrating how adaptations are conducted in practice (Escoffery et al., 2019; Leung et al., 2024; Fernández et al., 2025). However, it is crucial to integrate cultural adaptation research with the field of implementation science to better translate research findings into routine care settings, sustainably reach underrepresented communities, and promote health equity (Sangraula et al., 2021; Lau et al., 2023). Through a single-arm pilot of the adapted intervention in 2025, we aim to further our understanding of how the intervention might be continuously adapted and implemented within community-based settings.

Another limitation of the current study is that it is unclear whether the adaptations made to the intervention would necessarily meet its goals of improving youth mental health outcomes and fostering SEL and identity development. Given existing adaptation literature on cultural and ethnic invariance (Huey et al., 2014; Jones et al., 2018), it is critical to test the potential benefits of our adaptation. For example, what are the specific adaptations, such as SEL, identity development, and community-building for youth, that lead to more effective implementation and engagement? We also recommend comparing the effects of our adapted intervention with the original intervention through RCTs to explore whether our adaptations are causally related to improved mental health outcomes (Huey et al., 2014). Moreover, future research should also more critically examine the most effective adaptation strategies to meet the mental health needs of underrepresented groups (Park et al., 2023).

Further imitations of the current study include the small sample size and variations in adolescent and caregiver attendance. Due to scheduling limitations and technological difficulties, we did not always have the same number of participants in all the FGDs, and though we opened up the FGDs to newcomers to ensure we had a good number of participants in each session, the sample size was still limited to 18 adolescents and 12 caregivers. The small sample size may limit the generalizability of findings to other communities within or outside the US context. Despite these unavoidable logistical limitations, our sample size was consistent with similar adaptation studies (Garabiles et al., 2019; Sit et al., 2020; Tian et al., 2023), and we obtained valuable insights into the adaptation process. However, future work with large samples will be important for understanding the kinds of adaptations that are most important.

Finally, the study did not formally address saturation. Saturation refers to the point in qualitative data collection when no new, relevant information is found (Busetto et al., 2020; Johnson et al., 2020; Hennink and Kaiser, 2022). Saturation is typically assessed through counting the number of codes across transcripts until few or no more

codes are identified (Hennink and Kaiser, 2022). Although we had an iterative process of reviewing notes from diverse data sources, discussing observational feedback and interview responses, and reaching a consensus on our results during regular team meetings, we did not establish a stopping criterion for saturation. This was primarily because we had a limited number of FGDs and participants. Despite this limitation, our sample size fell within the recommended range of interviews and FGDs for saturation (Namey et al., 2016; Hennink and Kaiser, 2022), indicating that our findings captured the depth of issues studied (Francis et al., 2009).

**Open peer review.** To view the open peer review materials for this article, please visit http://doi.org/10.1017/gmh.2025.10045.

**Supplementary material.** The supplementary material for this article can be found at http://doi.org/10.1017/gmh.2025.10045.

**Data availability statement.** The authors confirm that the data supporting the findings of this study are available within the article and its Supplementary Materials. The changes made to the EASE intervention as documented in this study are available from the corresponding authors, A.D.B. and J.W., upon reasonable request.

**Acknowledgements.** The authors would like to thank the following three CBOs and their affiliated members for their partnership and support in the adaptation FGDs through May to June 2024: El Puente, a Brooklyn and Puerto Rico-based center for adolescents organizing, activism and community-led action (El Puente caregiver: Celeste Liriano); The BRO (Brothers Redefining Opportunity) Experience, a safe space for young men of color to express their feelings, share ideas and cultivate character in Bedford-Stuyvesant, Brooklyn; and the Center for Community Alternatives (CCA) Seeds to Roots Youth Action Center, located in Brownsville, Brooklyn. CCA supports and builds power with people across New York State who have been affected by mass incarceration, criminalization and community disinvestment (EASE CCA Staff: Rhamgurav Robinson, Senior Director of Community Programs; Chenequa Rogers, Career Readiness Coordinator; and Christine Song, Operations Coordinator).

**Author contribution.** Conceptualization: J.W., T.X., J.C., N.G.I., A.D.B.; methodology: J.W., T.X., J.C., N.G.I., B.A.K., A.D.B.; formal analyses: J.W., T.X., J.C., A.D.B.; investigation: J.W., T.X., C.S., J.C., N.G.I., D.E.S., A.D.B.; resources: J.W., T.X., C.S., J.C., N.G.I., E.W., H.D., K.G., E.A.; data management: J.W., T.X., C.S., J.C., N.G.I.; writing – original draft: J.W., T.X.; writing – review and editing: C.S., L.M., J.C., N.G.I., K.P., D.E.S., A.H., E.W., H.D., K.G., E.A., B.A.K., A.D.B.; visualization: C.S., L.M.; supervision: B.A.K., A.D.B.; project administration: J.W., T.X., E.W., H.D., K.G., E.A., A.D.B.; funding acquisition: A.D.B.

**Financial support.** The adaptation of EASE is funded by a grant from the New York City Mayor's Office of Community Mental Health (PI: A.D.B.).

**Competing interests.** All authors declare none.

**Ethics statement.** The New School University Institutional Review Board deemed this study Exempt from Human Subjects Research. However, all participating caregivers provided consent, while all participating adolescents provided minor assent. The caregivers of the participating adolescents also provided consent. The research team had resources available for participants should they be in need of referrals to manage distress. However, no referrals were needed in the current adaptation study.

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
