## [Reviewer Report]

I appreciate the authors’ use of a Community-Based Participatory Research (CBPR) approach, which is well described and grounded in the literature. This approach strengthens the manuscript and aligns well with contemporary best practices in community-based intervention research. However, to enhance clarity, coherence, and scientific rigor, I recommend addressing the points outlined below. Strengthening the structure, refining the discussion, and ensuring consistency between sections will greatly improve the manuscript’s quality and impact.

1. Introduction:

• The aim of the study should be clearly stated. While the manuscript provides background and context, the specific objectives should be explicitly outlined to help the reader understand the focus of the study.

2. Methods:

• The manuscript refers to caregivers, but it does not specify who they are. The authors should define caregivers explicitly, including their relationship to the adolescents (e.g., parents, guardians, extended family, or others). Clarifying this will enhance the reader’s understanding of the sample population.

• The acronym FGDs appears in the text, and I assume it stands for Focus Group Discussions. However, the first instance of this term should include the full phrase followed by the acronym in parentheses to ensure clarity.

• I appreciate the structured phased approach presented in the Methods section. However, this structure is not maintained in the Results section, making it difficult to follow the progression of the study.

3. Results:

• The Results section should mirror the structured phases used in the Methods section. This consistency will help readers follow the study’s findings logically and improve the clarity of the narrative.

4. Discussion:

• The opening of the Discussion states that the cultural adaptation process was an important step toward implementation. While this is true, it is neither novel nor well-argued in the text. The authors should better justify this point by connecting it to their findings or relevant literature.

• The section discussing future research directions at the end of discussion is somewhat vague. The authors should provide more specific recommendations for future studies.

5. Limitations:

• The limitations section should go beyond merely stating constraints. The authors should discuss the impact of these limitations on the study and justify their methodological choices. A more reflective and supportive discussion will strengthen this section.

---

## [Reviewer Report]

Community-based adaptation of Early Adolescent Skills for Emotions for Urban Adolescents and Caregivers in New York City

Thank you for the opportunity to review this important article. Providing culturally-responsive mental health programs is an important consideration to support the mental health and well-being of adolescents. This article provides an important example of a systematic approach to adapting existing evidence-based mental health programs to new contexts and populations. Moreover, the participants’ insights into culturally-relevant mental health terms and health promoting activities will be useful for other mental health programs that serve these cultures. The article illustrates the importance of exploring the cultural responsivity of a program before implementation and program evaluation as a “critical first step towards successful implementation.”

I believe the article would benefit from some restructuring to contribute to clarity. There are also a few places where clarifying information would be helpful. To make room for these additions, there are some redundancies throughout the paper that could be removed (e.g., between results and discussion section).

Some areas and questions to consider before publication:

- I strongly recommend reviewing APA guidelines for writing about Racial and Ethnic identities to reconsider some of the terms used in the article.

- Consider providing more information the cultural groups to whom the program will be offered in Brooklyn and how the focus group participants were identified for participation.

- In procedures, provide a brief description of the mhCACI framework so that the reader can evaluate your adherence to the framework. Understanding the rationale and sequence for each step would help readers understand the following descriptions of each step.

- For Phase 1, how did the first author and PI determine the core components? Did they review extant program evaluations conducted in other contexts?

- I found it a bit confusing that the in-depth literature review takes place in Step 2 to identify the program of interest after you have determined the programs core components. Having a clearer description of the mhCACI framework beforehand might help with this confusion.

- Participant Characteristics: Are the racial identity terms utilized those identified by participants (e.g., do participants refer to themselves as American Indians)? Do the participants represent the cultural groups who the EASE program will serve? What are the demographic characteristic of the parents/caregivers who took part in the focus groups?

- Train the trainers – had the research team completed a train-the-trainer training for the EASE Program? What role did they play given that the training was labelled as “self-guided.” Who were trained as facilitators and what are their demopgrahics? I found it a bit confusing that PM+ trainers were referenced here. What is the relationship of PM+ to EASE?

- 4. Translation of manual – what type of English was the manual written in? Did the language and terminology make sense to participants? Was any wording changed so that it was more relevant to the context in which the program was being offered?

- Results

o Consider providing a brief definition of each RECAPT criteria for unfamiliar readers, for example cultural concepts of distress.

o Re: cultural concepts of distress, I believe you are looking for how different cultures conceptualize distress. I’m not sure “a lack of understanding of distressing emotions” makes sense here, especially since it sounds like the adolescents came up with 50 terms describing psychological distress if I understood correctly. Did you mean that the vocabulary used in the original program did not align with participants’ cultural concepts of distress? As you go into the next section, it seems like participants have a lot of clarity on what distress looks like in their cultures.

o The choice of new additions to the program based on adolescent feedback would be strengthened by adding citations that support the efficacy for these strategies for mental health promotion. This would also apply to the sections on SEL and community building. What evidence in the literature indicates that the activities they choose promote mental health, social and emotional competencies and/or community building in adolescents?

o I found it a bit difficult to go back and forth between adolescents’ and caregiver feedback. Consider separating results. Also, were caregivers providing feedback on the activities for adolescents or were there separate caregiver materials? More clarity on this throughout the paper would be helpful.

o For activities that were ranked, is there a short, one-sentence description of what the activity is somewhere? Maybe it could be added to the table in which the results were presented?

o It would be helpful to introduce some procedures from qualitative research to strengthen study design (e.g., saturation, trustworthiness).